# Two-Stage Model-Based Predicting PV Generation with the Conjugation of IoT Sensor Data

**DOI:** 10.3390/s23229178

**Published:** 2023-11-14

**Authors:** Youngju Heo, Jangkyum Kim, Seong Gon Choi

**Affiliations:** 1DGB Financial Holding Company, Seoul 04521, Republic of Korea; youngju@dgbfn.com; 2Department of Data Science, Sejong University, Seoul 05006, Republic of Korea; 3School of Information and Communication Engineering, Chungbuk University, Cheongju 28644, Republic of Korea

**Keywords:** photovoltaic energy, IoT sensor, machine learning, ensemble model, data management

## Abstract

This paper proposes a novel short-term photovoltaic voltage (PV) prediction scheme using IoT sensor data with the two-stage neural network model. It is efficient to use environmental data provided by the meteorological agency to predict future PV generation. However, such environmental data represent the average value of the wide area, and there is a limitation in detecting environmental changes in the specific area where the solar panel is installed. In order to solve such issues, it is essential to establish IoT sensor data to detect environmental changes in the specific area. However, most conventional research focuses only on the efficiency of IoT sensor data without taking into account the timing of data acquisition from the sensors. In real-world scenarios, IoT sensor data is not available precisely when needed for predictions. Therefore, it is necessary to predict the IoT data first and then use it to forecast PV generation. In this paper, we propose a two-stage model to achieve high-accuracy prediction results. In the first stage, we use predicted environmental data to access IoT sensor data in the desired future time point. In the second stage, the predicted IoT sensors and environmental data are used to predict PV generation. Here, we determine the appropriate prediction scheme at each stage by analyzing the model characteristics to increase prediction accuracy. In addition, we show that the proposed prediction scheme could increase prediction accuracy by more than 12% compared to the baseline scheme that only uses a meteorological agency to predict PV generation.

## 1. Introduction

With the Paris Climate Agreement declaration in 2016, countries must set their own goals to reduce greenhouse gas emissions [1]. Such a trend has increased the importance of renewable energy generators and reduced fossil fuel consumption [2,3]. Since the PV generator has the advantage of not generating noise or pollution, they are issued as a significant alternative power resource. Furthermore, it is advantageous to install a PV generator in the downtown area so that the capacity size of the PV generator can be determined considering user preference. However, the PV generator has a problem because it is generation-sensitive to environmental change and seasonal factors [4]. The fact that the generator responds sensitively means that it is hard to predict power generation accurately, which means that the system operator could install incorrect generation capacity and induce various problems.

As a representative example, the uncertainty of renewable energy could occur in a rolling blackout event [5]. Such a problem could occur when renewable energy generation is lower than the expected value due to certain environmental changes (e.g., cloudy, rainy, etc). To prevent such issues, power generation companies secure additional reserve power in the system [6,7]. In an independent microgrid system (i.e., systemically independent power supply environment), the overproduction of renewable energy could cause an unstable issue in the system [8]. In this case, the system operator has to discard surplus power or transfer it into the main grid using the High Voltage Direct Current (HVDC) scheme [9].

To solve the uncertainty issues, controllable power sources (e.g., energy storage system, demand response) are introduced [10,11,12,13]. In [10], Zeynali et al. proposed a stochastic home energy management strategy to minimize overall monetary costs considering uncertainty in PV generation. The paper showed that the proposed power operation scheme was robust in various electricity tariffs and uncertain events minimizing overall cost. In [11], Sahbasadat et al. proposed a price-based demand response (DR) for industrial and commercial loads. The authors proposed an ESS operation scheme in the DR program to overcome the uncertainty issue to minimize the overall cost. Furukakoi et al. proposed a multi-objective genetic algorithm to improve voltage stability with the collaboration of the DR program and ESS [12]. By constructing the IEEE-6 bus system as a simulation environment, the authors showed that the proposed solution is efficient in the actual power system. In this way, various kinds of research show schemes that adopt additional power resources and ancillary services to solve the prediction error of renewable energy. However, such approaches could not be a fundamental solution when uncertainty exceeds the controllable range.

Therefore, various studies have been conducted to predict the amount of PV generation more accurately by adopting neural network-based prediction schemes [14,15,16]. In [17], Agga et al. tried to achieve accurate PV prediction results by merging two deep learning architectures, such as the CNN-LSTM model. The authors suggested that the proposed CNN-LSTM model showed excellent results compared to conventional prediction methods (i.e., LR, KNN, DTR, CNN, LSTM, and MLP). In [18], Dong et al. proposed a novel radial basis function to predict PV cell temperature. After predicting the cell temperature, the authors tried to combine the prediction results with the equivalent circuit model to predict PV generation. Lie et al. proposed a hybrid forecasting model using an enhanced gray wolf optimization algorithm to predict short-term PV generation [19]. By showing better convergence stability and higher accuracy compared with the conventional methods, the authors showed the distinction of the proposed scheme in the paper. In this way, various papers proposed novelty in terms of prediction schemes. However, the time unit of data needs to be considered, which is essential to implement the proposed scheme in the actual environment. Therefore, we propose a prediction technique considering the data acquisition time so that the development results can be used in the actual environment.

The remainder of this study is organized as follows. In Section 3, we introduce the overall system model and provide data processing details, which are essential to predict PV generation. In Section 4. We show various numerical results to demonstrate the suitability of the proposed scheme. Finally, we conclude this paper in Section 5.

## 2. Related Work

Currently, PV generation takes a large portion of energy supplement in the power system. It is advantageous in terms of environmental friendliness, less emissions, low noise, and space efficiency so it is installed easily in the system. However, there are problems associated with PV such that prediction is difficult due to environmental sensitivity issues [20]. Since PV is sensitive to environmental changes (e.g., cloud movement, solar radiation reduction), various approaches are proposed to increase the prediction accuracy of PV generators [21,22,23]. Using the weather data from the meteorological administration, Liu et al. proposed a PV prediction scheme using the weather cluster analysis and uncertainty estimation in [22]. In the paper, the authors show that uncertainty estimation makes it possible to achieve robust prediction results of PV generation and it is efficient to reduce prediction error in the system. In [21], Song et al. show the PV prediction scheme that selects three weather stations that are physically close or highly related to the target point and predict PV generation using the corresponding data. By proposing a partial daily pattern prediction framework, Wang et al. showed that it is possible to improve the accuracy of the day ahead PV prediction. Based on the ensemble technique of pattern classification and PV prediction models, the authors showed that the proposed model was capable of deriving higher accuracy than conventional models. However, there are limitations in previous research in that they cannot monitor sensitive environmental changes in the local area where solar panels are installed. As a way to solve such an issue, studies are being conducted to collect local data using various IoT sensors to increase the prediction accuracy of PV generation.

Using IoT sensors, it is possible to monitor sensitive environmental changes in the target area where solar panels are installed which is more related to PV generation. In [24], Kim et al. showed the PV prediction scheme using the IoT sensors gathered from the local site. By applying the statistical approach, the authors show the significance of the input variables and construct a model to predict PV generation using the selected variables. In [25], Ahn et al. showed a short-term PV prediction scheme using a deep-RNN model using power from the Internet of Things devices. By analyzing numerical simulation results that compare prediction results between the proposed method and conventional methods, the authors showed that it is efficient to use various IoT sensors to predict PV generation. In this way, various studies are being conducted to use various types of IoT sensors to sensitively detect environmental changes in the region and maximize the prediction accuracy of PV prediction. However, conventional studies have limitations in that they exclude consideration of the time of data acquisition. Generally, when we use IoT sensors, it is hard to achieve sensor data at the time we prefer to predict PV generation. Therefore, we propose a method of predicting PV generation by sequentially predicting the sensor data at that point in time and then using it in the next stage.

In this light, the main contributions of this paper are summarized as follows.

In various conventional research studies, PV generation predicts using average environmental data in a large region [26]. In this case, there is a limitation in that it is challenging to detect the environmental change in the local area where the solar panels are located. To solve such a problem, we install IoT sensors (e.g., temperature sensor, irradiation sensor) near the generator to monitor environmental changes in the area and propose a proper prediction scheme in the given environment.To predict PV generation using IoT sensors, we have to consider the point of time that data achieves. In general, the data of the required IoT sensor is achieved after we predict the PV generation. Therefore, we propose a two-stage prediction scheme that could be used in an actual power system. In the first stage, we predict future IoT sensor data based on accumulated environmental data. In the next stage, PV generation predicts using the data achieved in the previous stage.In the proposed scheme, prediction performance can vary depending on the type of neural network model that constitutes each stage. To solve such an issue, we are able to derive a combination that shows optimal performance by combining various types of neural network models and prove the effectiveness of the proposed method through performance comparison with conventional schemes.In this paper, we install two types of IoT sensors (i.e., temperature sensor and solar irradiation sensor) located near solar panels. Although there are various sensor data, some have overlapping meanings, making it hard to achieve accurate prediction results. To solve such issues, we analyze the correlation and multicollinearity of the data set and determine an appropriate PV prediction scheme to achieve high prediction accuracy.

## 3. System Model

As depicted in Figure 1, two types of data are used to predict PV generation in a particular area. The first one is IoT sensor data which detects an environmental change in the solar panel area. The other is estimated environmental data which refers to a wide range of average environmental information announced by meteorological administration. Since IoT sensor data are collected from the sensors installed on the rooftop of the building where the solar panel is located, they contain the latest information about the surrounding environment (i.e., cloud position, temperature, dust, etc.) that are the most vital to predicting PV generation. However, the data acquisition could only be performed after PV generation, and using IoT sensor data in a real-time mechanism is difficult. On the other hand, environmental data can achieve specific periods before (e.g., a day ahead, 12 h ahead, etc.) provided by meteorological administration. Here, environmental data have the advantage of providing future information at the point where we try to predict PV generation. However, environmental data mean regional average and need to obtain additional detailed regional information.

In this way, each dataset has different information and limitations, making it hard to use directly. Therefore, we propose a two-stage PV prediction scheme that reflects each dataset’s characteristics. In Figure 2, we show the details of the proposed prediction scheme using the given data set. Here, seven types of environmental data are used to predict IoT sensors. Since IoT sensor data contains information on the area where the solar panel is located, we assume that predicted data will be directly related to PV generation. Therefore, we analyze the relationship between IoT sensors and PV generation with various correlation techniques and determine an appropriate prediction model to achieve accurate results.

### 3.1. Data Analysis

Before implementing prediction models, we try to select a suitable candidate neural network model based on the analysis of the given data set. By analyzing the relation between the data, we decide whether to use the given data set to construct the model by analyzing the linearity.

#### 3.1.1. Correlation Analysis

As argued in the above section, the proposed PV prediction process is divided into two stages. In the first stage, we predict IoT sensor data using an environmental data set. Then, PV prediction is progressed using the predicted IoT sensor data achieved in the first stage and estimated environmental data. Before calculating prediction results through the neural network model, we analyze the relation between input and output data at each stage to determine an appropriate prediction model. Table 1 shows the analysis results for correlation in the first stage.

In Table 1, we use three representative methods (i.e., *Pearson*, *Spearman*, and *Kendall*) to analyze the correlation between the data set. Generally, *Pearson correlation* is used to measure linearity between two variables. Here, *Pearson correlation* equal to 1 refers to the fact that there is a strong positive relationship between the data, and −1 means that they have a strong negative relationship. In addition, when *Pearson correlation* is close to 0, two variables have no linear correlation. *Spearman correlation* is used to evaluate the monotonicity relation between data sets. Here, monotonicity means that one variable’s magnitude changes affect another variable’s exact magnitude change. When the value of *Spearman correlation* is close to 1, two variables have a monotonic correlation. On the other hand, the correlation value becomes 0 means variables have less monotonic correlation. *Kendall correlation* is similar to the *Spearman correlation*, used to judge a data set’s monotonicity. Generally, we can confirm a monotonic relation between datasets when there is a linear relationship. However, the monotonic relationship does not imply that there must be a linear relationship between data sets. Therefore, it is necessary to analyze correlation using various methods.

In the table, we can check the correlation between IoT sensor data and environmental data using various correlation schemes. Since irradiation refers to the amount of isolation that reaches the earth’s surface, we could guess that it has a high relation with cloud and humidity. This relationship can be proven through the correlation analysis of data (i.e., the value of *Pearson correlation* for Vertical irradiation and Humidity is −0.397 and for Vertical irradiation and Cloud is −0.335). Also, we could check that the correlation value between the temperature data of the IoT sensor and the pressure from environmental data is high (i.e., the value of *Pearson correlation* for Module temperature and Pressure is −0.733). We could check this kind of close relationship occurs due to the Universal Gas Law theorem. Lastly, the temperature has a high correlation with the visibility (i.e., the value of *Pearson correlation* for module temperature and visibility is 0.322 and outdoor temperature and visibility is 0.318). Due to the change in the altitude of the atmospheric boundary layer, we determine that it is accurate to have a correlation between the two data. With such knowledge of geoengineering and correlation analysis results, we could determine that some of the data have a linear relationship, and some can be presumed to have a non-linear relationship. Therefore, we select a prediction model in each stage considering such relation among the data set.

Table 2 deals with the relationship between IoT sensor data and PV generation. In the table, we use *Pearson*, *Spearman* and *Kendall* technique to analyze the correlation [18,19]. Through analyzing correlation results, we can check that vertical irradiation and horizontal irradiation have a strong linear relationship with PV generation, and PV module temperature and outdoor temperature also have a particular relationship with PV generation. In addition, it confirms that similar results appeared in other indicators, which analyze monotonicity and linearity. Therefore, we could confirm that the data achieved through the IoT sensors closely relates to PV generation.

Since similarity is also one of the important factors in analyzing the relation between the data set, we calculate the dynamic time warping (DTW) value of data. Here, DTW is an algorithm that measures the similarity of two time series data with different movements. Since IoT sensors and PV generation are time series data that have continuity according to period, we assume that the DTW technique would be efficient in analyzing the similarity of the dataset. Using the DTW scheme, we could determine that vertical and horizontal irradiation of data have a higher relation with PV generation as depicted in Table 3.

#### 3.1.2. Multicollinearity Analysis in Prediction Model

As checked in Table 1, it shows that the parts of environmental data and IoT sensor data have a linear relationship. Since a linear-based regression model could be one of the solutions, it is essential to perform multi-colinearity testing on the data set to select an appropriate predictive model.

There are two representative methods for confirming multicollinearity in the dataset: (i) check multicollinearity using a scatter plot graph; (ii) verification by calculating Variance Inflation Factors (VIF). Since the verification of multicollinearity through a scatter plot graph has a subjective opinion of the researcher, we try to verify it through the calculation of VIF in this paper. Generally, VIF is a measure to check that there is a correlation between independent variables in a multiple regression model. When the value of VIF is around 1, there is no multicollinearity among the data, and the model is reliable. However, if VIF is greater than 10, variable selection should carefully occur because the data set has high multicollinearity. The detailed formula of the VIF index could be described as follows:(1)$$\begin{array}{cc} VIF_{i} & =\frac{1}{1-R_{i}^{2}}. \end{array}$$

In Equation (Equation 1), $VIF_{i}$ refers to the VIF value of the *i*th variable. Here, $R_{i}^{2}$ is the coefficient of determination which indicates the change in the response variable according to the predictor variable. By applying the VIF index formula, we could achieve VIF values for environmental and IoT sensor data as shown in Table 4 and Table 5.

In Table 4 and Table 5, we can check humidity and pressure in the environmental data set, vertical irradiation and horizontal irradiation, and PV module temperature and outdoor temperature in IoT sensor data have high multicollinearity. Generally, the most straightforward way to solve this issue is to delete part of the data with high VIF so that the VIF value for all data becomes lower than 10. However, the usable data set will be reduced in this approach. Therefore, we achieve prediction results by implementing a prediction model that could evade multicollinearity issues.

### 3.2. Two-Stage PV Prediction Scheme

As mentioned in the above sections, we try to predict the IoT sensor data using environmental data first and achieve the PV prediction results through the predicted sensor data. However, the issue of multicollinearity exists in some of the collected data, which could cause several issues in prediction progress. To solve such an issue, we propose various 2-stage PV prediction schemes as depicted in Figure 3.

In the figure, we propose various schemes to predict PV generation. Here, Figure 3a expresses the scheme discussed in conventional papers that predicts PV generation using only environmental data provided by the meteorological administration. Moreover, Figure 3b describes a method of creating various models so as not to duplicate information with the consideration of the multi-colinearity between the data. In this scheme, a weighted sum of each prediction result achieves PV generation. Finally, Figure 3c covers the scheme to use the whole dataset in each stage. In this scheme, to avoid multi-collinearity issues, we select the following candidate model to achieve.

#### 3.2.1. Long Short-Term Memory

Long short-term memory (LSTM) is a recurrent neural network (RNN) model that is frequently used to predict time series data. Since LSTM has a memory in the recurrent neural network, it has a feature to catch pattern changes of past time series data [27,28,29]. In addition, unlike other models, it is easy to achieve prediction results by considering the order and timing of the data. Therefore, we determine that LSTM can exhibit high performance in predicting time series data and choose it as one of the options to predict IoT sensors and PV generation. The details of the model that we construct in this paper are as follows.

In Table 6, we describe the LSTM model used to predict IoT sensors. In the fully-connect layer, we use sigmoid as an activation function in the fully-connect layer since PV generation expects to move between specific bands. In addition, to remove the negative values, we put the ReLU function in the last of the FC layer.

#### 3.2.2. Extreme Gradient Boosting (XGboost)

Most papers argued that Extreme Gradient Boosting (XGboost) is the most appropriate scheme for analyzing time series data. XGboost is a decision tree-based ensemble model organized with a Classification and Regression Tree (CART). In general, a tree-based model has a limit in that it cannot predict a value out of boundary value among the given data set. However, we could judge that PV power to predict would not be located out of the boundary set. In addition, in many cases, XGBoost performs excellently predicting various power data [30,31]. Therefore, we choose XGboost as one of the models to predict PV generation.

In our paper, we construct the XGboost model by optimizing the main parameter using a grid search scheme. However, there are no specific parameters that show excellent results. Therefore, we set the value within the universal range.

## 4. Simulation Results

This section uses a proper neural network model to show prediction results such as PV generation and IoT sensor data. Here, we define three processes to achieve prediction results. First, we use various schemes to predict IoT sensor data and find optimal schemes with optimal parameters to increase prediction accuracy. In addition, we show prediction results of PV generation using the predicted data set in the first stage through the various neural network models. Finally, we achieve PV generation value, and the suitability of the proposed scheme is analyzed by comparing the results with the conventional methods.

In the whole simulation results, prediction results are derived using the data collected from a PV generator installed at the Korea Advanced Institute of Science and Technology (KAIST), environmental data collected from IoT sensors installed near generators, and meteorological administration in Daejeon. In the system, the PLC modem and sensor module installed in the solar panel are collected by the Data Concentration Unit (DCU) in the central control room of the university based on Ethernet communication and can be monitored through the SCADA network. The whole dataset is collected from 1 February to 31 December 2018. Here, we set the time unit of data on a 1-h basis from 5:00 am to 10:00 p.m. that PV generators operate. To validate the proposed prediction scheme, we use 80% of the data set as a training set and treat the remaining 20% of the data as a test set.

In order to evaluate the performance of the proposed prediction method through the sampled test set, the prediction error is measured using two metrics, such as Root Mean Square Error (RMSE) and Mean Absolute Error (MAE) [32]. When using other indicators that deal with the error probability according to the actual value, the percentage could increase in the section where the actual value is less than 1. Therefore, accuracy is analyzed with the corresponding techniques that analyze the error between the actual value and prediction values. Generally, RMSE and MAE are defined as
(2)$$RMSE=\sqrt{\frac{1}{N}\sum_{t=1}^{N}{\left(o_{t}-{\tilde{o}}_{t}\right)}^{2}},$$
(3)$$MAE=\frac{1}{N}\sum_{t=1}^{N}\left(\left|o_{t}-{\tilde{o}}_{t}\right|\right)$$
respectively. In Equations (Equation 2) and (Equation 3), *N* represents the number of daily sampling points. In addition, $o_{t}$ and ${\tilde{o}}_{t}$ refer to the actual and predicted outputs at time *t* of the day, respectively.

### 4.1. Data Pre-Processing

Before implementing an accurate prediction model, we need proper data pre-processing (i.e., synchronize time sequence of data, interpolate missing data) to achieve data for training. In the case of environmental data, it is organized with various time units from 0 to 24 h. However, the IoT sensor operates from 5:00 am to 9:00 pm In addition, the time slot of IoT sensor data is different (i.e., 30 min or 1 h) depending on the settings of the modules. In order to utilize the whole data set, we transform the time unit of the data on a 1-h basis. Here, data with a time unit of more than 1 h secured an insufficient value through the interpolation process [33], and data whose unit is shorter than 1 h took an average value to achieve the simultaneous time slot.

In the case of IoT sensor data, some of the collected data is missed or damaged due to various issues (i.e., low communication quality, external environment change, incorrect data processing). Since such erroneously collected data makes it hard to predict PV generation, we achieve high-quality data by applying an appropriate pre-processing scheme. In our paper, we achieve an average data set by applying a k-nearest neighbor algorithm-based interpolation scheme after detecting outliers to solve insufficient data issues [34,35].

### 4.2. IoT Sensor Prediction Result

This section deals with the IoT simulation results based on various neural network models. According to the various models mentioned in Section 3.2.1 and Section 3.2.2, we determine that the use of a recurrent neural network model and a tree-based model are most appropriate for predicting data considering the characteristics of the data itself. In order to prove this claim, we compare the prediction results for each model in Figure 4.

In the figure, we predict solar irradiation using two prediction methods. The figure confirms that achieving high prediction accuracy using an environmental data set is possible. However, in most cases, it can be seen that XGboost derives prediction results that are closer to actual data than the LSTM scheme. In the case of the tree model, there is a problem in that performance drops sharply when the prediction range is out of the given data set. However, considering the given sensor data, there is no occurrence of a situation in which the prediction progresses outside of range. Therefore, we could check that the accuracy of XGboost is higher than LSTM. This could be confirmed by constructing the models to predict various sensors as follows:

In Table 7, we show the prediction results of IoT sensors using various prediction schemes. From the table, we can confirm that XGboost, in whole cases, maximizes the IoT sensor’s prediction accuracy. In particular, it confirms that the performance of XGboost is excellent in the case of Vertical and Horizontal irradiation where a significant fluctuation of values exists. However, we could confirm that the difference in accuracy with LSTM is small in the case of temperature data. Considering such prediction accuracy, we plan to analyze the prediction results of PV generation using the predicted results.

### 4.3. Comparison of PV Prediction Accuracy by Proposed Methods

This section attempts to predict PV generation using collected environmental and IoT sensor data. As argued in the previous section, we use three prediction schemes with two different neural network models. Therefore, we aim to compare the prediction results based on the proposed methods (i.e., Offline (actual sensor data), Baseline (weather data only), 2-stage model (using the whole of predicted data), and Ensemble (partially used data in prediction)). Here, we experiment by changing the model used for each stage to analyze whether the prediction of sensor values affects the PV prediction value. Therefore, the simulation progresses as follows:

In Table 8, we can check the performance of various prediction schemes. *Offline* is the predicted result based on the actual sensing IoT sensor and weather data, and *Baseline* is the case that uses the most accurate model using the weather data only. While comparing the two results, sensor data significantly affects PV prediction accuracy. In other words, predicting IoT sensor data accurately is essential to achieve PV prediction. In addition, since the given models can derive prediction results while avoiding multiplicity problems, the *2-stage model* shows high accuracy compared with the *Ensemble* scheme, which uses data selectively in each model. Finally, the results achieved through the *2-stage model* show that using a neural network affects the prediction accuracy. Here, in order to maximize the accuracy of the proposed method, we select the most appropriate models by combining representative neural network models that are commonly used in predicting time series data. When predicting IoT sensor data, it has been confirmed that using the XGboost model produces stable prediction results. XGboost and BiLSTM models make suitable prediction results in predicting PV generation. The simulation confirms that the prediction accuracy can be improved by up to 12% using appropriate prediction approaches and models.

In Figure 5, we compare the prediction performances according to the different neural network models in the second stage. In the figure, we can see that it is difficult to predict PV generation at the peak point when using the LSTM series neural network models. In particular, it is confirmed that prediction values are significantly different from the actual values around 13:00 and 29:00 in the figure. In contrast, it is confirmed that the prediction results are actually higher when using the XGboost model. In particular, we can see the prediction error in the peak period is lower than the LSTM series model, which is an important factor to apply in the actual power industry.

### 4.4. Seasonal Prediction Analysis

There are two types of electricity tariffs in power systems: setting rates in advance based on power consumption and supplement patterns and charging costs according to real-time power supplements. A common issue that these pricing policies are related to is the amount of power supplement in the system. Since the variability of renewable energy in summer and winter is higher than in other periods, it is judged to predict more accurately than other periods. Therefore, we aim to show that the proposed model can contribute to reducing the prediction error in the corresponding period by analyzing the prediction results during the corresponding period in Figure 6.

In the figure, we check the prediction results in the summer and winter seasons in South Korea. Analyzing the prediction results during the summer period in Figure 6a, it is possible to confirm that predicting IoT sensors using the XGboost model improved prediction accuracy. Conversely, prediction results during the winter period in Figure 6b show that usage of IoT sensor data affects the prediction accuracy rather than determining the proper neural network model. Since there are many environmental changes in the region in winter, monitoring the environmental change in the region derives the predicted result of renewable energy.

## 5. Conclusions

In this paper, we proposed a two-stage PV prediction scheme using on-site IoT sensors. First, on-site IoT sensors predict using various neural network models with the environmental data provided by the meteorological agency. Next, the PV predictions progress by using the predicted IoT sensors and environmental data. With the various simulation results, the proposed two-stage model can reduce prediction error by 12% (MAE) and 2.4% (RMSE) compared to the case that achieves prediction results only using environmental data. In addition, by determining the appropriate neural network model in each stage, it is confirmed that the proposed method could maximize the prediction accuracy rather than using different prediction schemes or neural network models. The results confirm that sensitively recognizing environmental changes in the region significantly impacts PV prediction. Therefore, considering other factors, including real-time data or image recognition technology, we expect to achieve improved prediction results in future work.

## Figures and Tables

**Figure 1 sensors-23-09178-f001:**
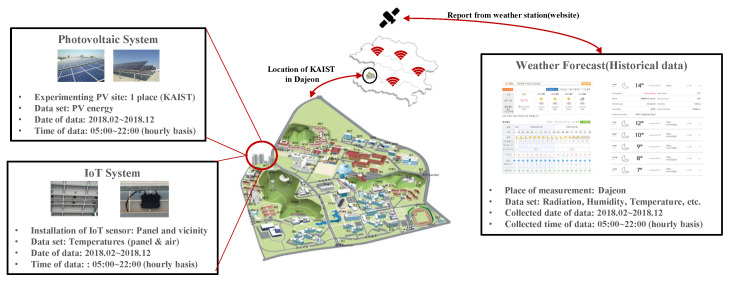
System model of the proposed PV prediction scheme.

**Figure 2 sensors-23-09178-f002:**
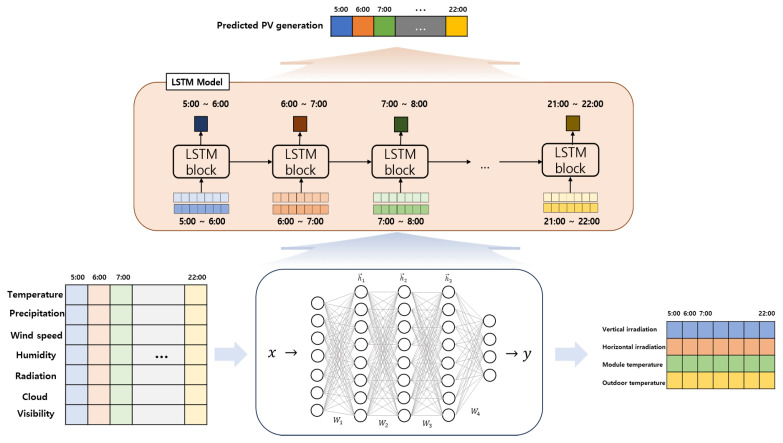
Proposed two stage PV prediction scheme.

**Figure 3 sensors-23-09178-f003:**
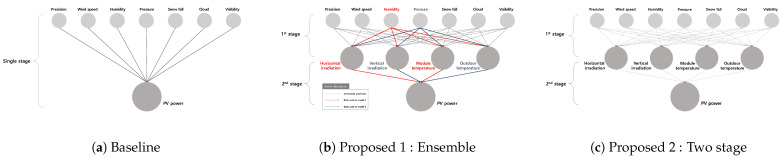
Various schemes to predict PV generation.

**Figure 4 sensors-23-09178-f004:**
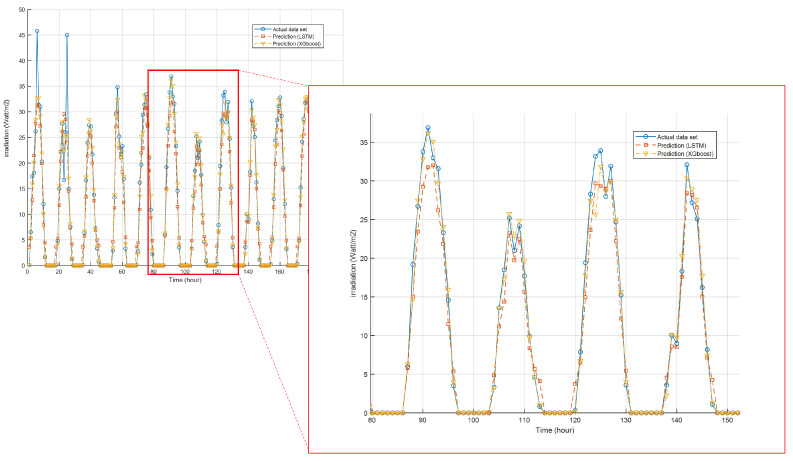
IoT sensor prediction results in the winter period.

**Figure 5 sensors-23-09178-f005:**
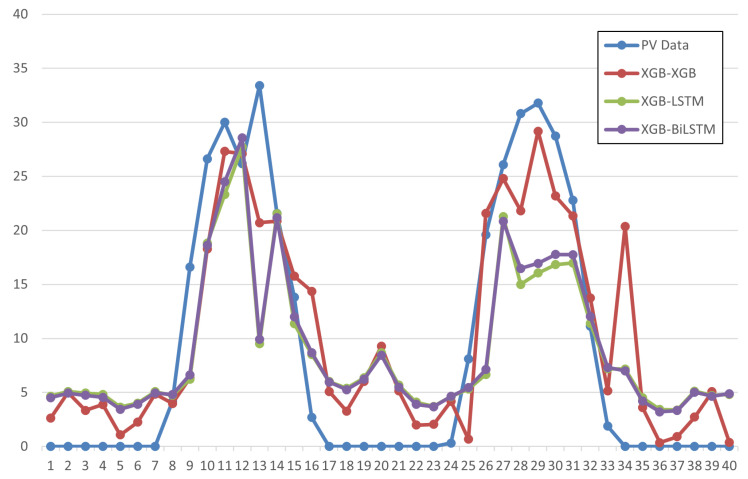
Comparison of prediction performance according to PV prediction models.

**Figure 6 sensors-23-09178-f006:**
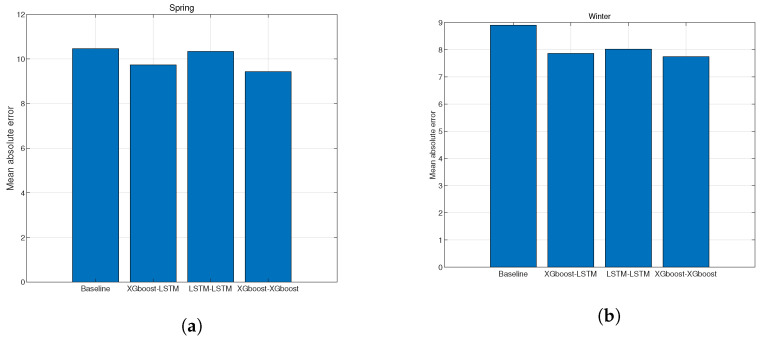
Comparing prediction accuracy using seasonal data: (**a**) prediction accuracy comparison using spring data; (**b**) prediction accuracy comparison using winter data.

**Table 1 sensors-23-09178-t001:** Correlation analysis in the first stage (i.e., between Environmental data and IoT sensor data).

IoT Data	Correlation	Precipitation	Wind	Humidity	Pressure	Snow	Cloud	Visibility
Vertical irradiation	Pearson correlation	−0.129	0.179	−0.397	−0.05	−0.007	−0.335	0.182
	Kendall correlation	−0.171	0.134	−0.277	−0.02	−0.003	−0.249	0.154
	Spearman correlation	−0.211	0.201	−0.401	−0.03	−0.004	−0.344	0.228
Horizontal irradiation	Pearson correlation	−0.133	0.165	−0.443	0.09	0.001	−0.409	0.187
	Kendall correlation	−0.181	0.129	−0.309	0.06	0.003	−0.298	0.154
	Spearman correlation	−0.223	0.193	−0.443	0.09	0.004	−0.405	0.229
Module temperature	Pearson correlation	0.014	0.138	−0.047	−0.733	−0.089	0.082	0.322
	Kendall correlation	0.002	0.099	−0.034	−0.533	−0.075	0.042	0.235
	Spearman correlation	0.002	0.146	−0.048	−0.740	−0.091	0.065	0.342
Outdoor temperature	Pearson correlation	−0.045	0.139	−0.217	−0.505	−0.074	−0.100	0.318
	Kendall correlation	−0.074	0.107	−0.143	−0.348	−0.066	−0.086	0.235
	Spearman correlation	−0.091	0.160	−0.216	−0.517	−0.081	−0.118	0.344

**Table 2 sensors-23-09178-t002:** Correlation analysis in the second stage (i.e., between PV generation and IoT sensor data).

	Correlation	Vertical Irradiation	Horizontal Irradiation	PV Module Temperature	Outdoor Temperature
Power generation	Pearson correlation	0.910	0.934	0.329	0.649
	Kendall correlation	0.892	0.905	0.276	0.504
	Spearman correlation	0.973	0.973	0.390	0.681

**Table 3 sensors-23-09178-t003:** Similarity analysis between PV generation and IoT sensor data.

	Similarity	Vertical Irradiation	Horizontal Irradiation	PV Module Temperature	Outdoor Temperature
Power generation	Dynamic Time Warping	9.297	10.557	38.058	28.901

**Table 4 sensors-23-09178-t004:** VIF analysis for environmental data.

Environmental Data	Precipitation	Wind	Humidity	Pressure	Snow	Cloud	Visibility
VIF factor	1.086	3.148	14.861	21.237	1.005	3.431	3.822

**Table 5 sensors-23-09178-t005:** VIF analysis for IoT sensor data.

IoT Data	Horizontal Irradiation	Vertical Irradiation	PV Module Temperature	Outdoor Temperature
VIF factor	65.751	62.262	84.854	157.269

**Table 6 sensors-23-09178-t006:** Configuration of the LSTM-based neural network.

Layer	Description
LSTM block	Input (Hourly meteorological information)
	Hidden layer (Number of input dimensions)
	Output State
FC layers	Input (Output state) (5)
	Linear Regression (Number of hidden layer -> output size: 1)
	Sigmoid
	ReLU
	Output (Hourly PV energy generation) (1)

**Table 7 sensors-23-09178-t007:** Prediction error of the IoT sensor data using various neural network models.

**(a)** Comparison of prediction error by MAE value				
	Vertical irradiation	Horizontal irradiation	Module temperature	Outdoor temperature
LSTM	6.9069	6.9026	6.9198	6.9186
XGboost	1.0397	1.7508	6.3432	5.0549
**(b)** Comparison of prediction error by RMSE value				
	Vertical irradiation	Horizontal irradiation	Module temperature	Outdoor temperature
LSTM	10.2299	10.2223	10.2525	10.2502
XGboost	3.1236	2.43	9.8152	8.3853

**Table 8 sensors-23-09178-t008:** Prediction errors of PV generation.

Approaches	IoT Prediction	PV Prediction	MAE	RMSE
**Offline**	**Real Data**	**LSTM**	**2.61**	**3.07**
Baseline	-	LSTM	9.04	10.51
2-stage	LSTM	LSTM	8.62	10.41
Ensemble	LSTM	LSTM	9.50	11.28
Baseline	-	BiLSTM	9.30	11.40
2-stage	BiLSTM	BiLSTM	8.58	10.38
Enseble	BiLSTM	BiLSTM	8.85	10.38
Baseline	-	XGboost	8.17	10.34
**2-stage**	**XGboost**	**XGboost**	**8.06**	**10.36**
Ensemble	XGboost	XGboost	8.19	10.14
2-stage	LSTM	XGboost	8.76	10.66
2-stage	XGboost	LSTM	8.22	10.17
2-stage	BiLSTM	XGboost	8.56	10.36
**2-stage**	**XGboost**	**BiLSTM**	**8.08**	**10.03**

## Data Availability

Data are contained within the article.

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
