# Peer review of "Two-Stage Model-Based Predicting PV Generation with the Conjugation of IoT Sensor Data"

_sensors, 2023, doi:10.3390/s23229178_

Round 1

Reviewer 1 Report

Comments and Suggestions for Authors

In this paper, the authors propose a short-term photovoltaic voltage prediction scheme using IoT sensor data with a two-stage neural network model. The approach leverages environmental data from meteorological agencies to predict future PV generation. While the paper presents an interesting concept, there are several areas that require improvement:

(1) The abstract should clarify the specific motivation behind designing the "two-stage neural network model." Many existing works already employ multi-stage models, so the novelty in this regard needs to be highlighted.

(2) Figures in the paper do not meet journal requirements, primarily due to low-resolution images and small caption sizes. Enhance figure quality for better readability and comprehension.

(3) The comparative analysis is relatively simplistic compared to recent forecasting works. The paper mainly compares the proposed method to classical methods. To demonstrate the effectiveness of the approach, consider comparing it with advanced forecasting models commonly used in recent research.

(4) In Section 3.1, the data collection scheme's details are vague, particularly regarding how IoT frameworks are used to obtain the data. Provide a clearer explanation of the data collection process.

(5) Improve the language quality and address grammar mistakes, such as the phrase "Table 6 show."

(6) Clarify the rationale for using LSTM in the proposed method, as there are various neural network architectures available. Explain why LSTM was chosen over other alternatives.

(7) If the paper explores seasonal prediction analysis, it should include comparisons with Seasonal Autoregressive Integrated Moving Average (SARIMA) models for a more comprehensive evaluation.

(8) The paper structure could be enhanced with more subsections and a more comprehensive literature review. Address the writing logic in the literature review section and include recent forecasting models that are currently missing.

(9) Figure 2 indicates that many enhanced LSTM models use a similar structure, potentially indicating a lack of theoretical advancement. Clarify the unique contributions of the proposed model compared to existing LSTM-based models.

Comments on the Quality of English Language

Moderate editing of English language required.

Author Response

Dear reviewer, 

I am really appreciate on your suggestion. On the revised version of paper, we add various simulation results and rewrite the paper to enhance the quality of paper. 

The action for your comments are described in the attached response letter.

Thank you for your help 

Jangkyum Kim

Reviewer 2 Report

Comments and Suggestions for Authors

·        Figures should be placed after the citation in the text.

·        Show the type of IoT sensor, and describe used communication protocols for transfer and where the data are stored.

·        Is this a real or virtual system?

·        What do you mean by "vertical" and "horizontal" irradiation in Table 1?  Do you measure global, direct, and diffuse irradiation? Module

·        temperature should be PV module temperature.

·        Please provide more analysis of Table 1.

·        The caption should be above the table.

·        In section 2.1.1. needs revision - analysis of the data from Table 1. The first paragraph on page 5 analyses data from other papers. For example, irradiation refers to the amount of isolation that reaches the earth's surface; the relationship between temperature and pressure according to the Universal Gas Law - Is this for PV module temperature? According to the data, the Person correlation between PV module temperature and air pressure is -0.733, and the Spearman correlation is - 0.74. Is this true?

·        How you obtain this information from your IoT system - temperature change is also closely related to visibility because it induces a change in the altitude of the atmospheric boundary layer.

·        Please revise those sentences "With such knowledge of geoengineering and correlation analysis results, we could confirm that environmental data has a high relationship with IoT sensor data and can be used to predict IoT sensor value. Here, it confirms that there is no clear linear relationship." In my opinion, for -0.733 we have a good correlation.

·        Dynamic time wrapping should be Dynamic Time Warping (DTW)  - Warping. and this is not a correlation, so this parameter is not suitable to be in Table 2.

·        There is a mistake in the row 245.

·        Used symbols in the equations and text must be the same.

·        Extend the conclusion section with the main results from the paper.

·        The monitoring parameters are 4: Vertical irradiation, Horizontal irradiation, Module temperature, and Outdoor temperature.

Comments on the Quality of English Language

Please check the manuscript for grammatical errors.

Author Response

(The authors gave the same response as above.)

Reviewer 3 Report

Comments and Suggestions for Authors

The manuscript titled "Two-Stage Model for Predicting PV Generation with IoT Sensor Data" proposes an innovative two-stage neural network model for forecasting short-term photovoltaic (PV) voltage using IoT sensor data. The aim is to address the limitations of relying on average environmental data from meteorological agencies, which may not accurately represent specific installation areas for solar panels. To overcome this, the authors propose using IoT sensor data to detect environmental changes in the specific area. The two-stage model involves utilizing predicted environmental data to access IoT sensor data in the desired future and then predicting PV generation using the predicted IoT sensor and environmental data. The authors demonstrate that their proposed scheme achieves higher forecast accuracy, outperforming the baseline scheme that relies solely on meteorological agency data by over 12%. It can be concluded that this study is well-written.

While the manuscript appears to be well-written, there are a few minor points that could enhance its quality. Firstly, the abstract should concisely and comprehensively represent the study's findings. Additionally, it is recommended to include a brief discussion on machine learning applications and algorithms in the introduction, supported by relevant references such as "Deep learning: Applications, architectures, models, tools, and frameworks: A comprehensive survey,"

"Efficient framework for detection of COVID-19 Omicron and delta variants based on two intelligent phases of CNN models," and

"A fast and efficient CNN model for B-ALL diagnosis and its subtypes classification using peripheral blood smear images."

The topic of this research is original and relevant to the field of PV power prediction. Most prior work on PV prediction has relied solely on broad regional weather forecasts. This research proposes leveraging real-time IoT sensor measurements near the PV installation, which is not commonly done. This adds a localized environmental data element that can improve accuracy.

The subject area compared with other published material
Use of on-site IoT sensor data: Most prior work only used regional weather forecasts, not localized real-time sensor measurements near the PV installation. This adds a new data dimension not fully leveraged before.

Two-stage predictive model: By predicting sensor values first before PV output, it addresses the gap of not being able to directly fuse forecast and sensor data due to timing mismatches. This novel two-step approach is absent from previous work.

Correlation and multicollinearity analysis: It performs in-depth analysis of the relationships between different data sources to optimize variable selection for the models. This level of data preprocessing is more thorough.

Evaluation of multiple neural networks: It compares the performance of LSTM, XGBoost and an ensemble approach for each prediction stage, providing insights on optimal algorithms. Most studies only used one technique.

Main questions addressed by the research

How can authors leverage IoT sensor data to better detect local environmental changes affecting the PV panels, as opposed to just using broad regional weather data?

How can authors structure a prediction model to utilize both future forecast data and real-time sensor data, given that the sensors report after the target prediction time?

What is the optimal way to select and combine the different data sources while addressing issues like multicollinearity between some variables?

Can authors' proposed two-stage neural network approach improve PV generation prediction accuracy compared to just using weather data or other existing methods?

Specific improvements the authors should consider regarding the
methodology. Further controls should be considered.
Expand sensor types/placements - Test adding other locally relevant sensors like wind speed, humidity, etc. More/different sensor placements could provide additional useful data.

Increase training data size - The sample size of 1 year of data is relatively small. Using a larger multi-year dataset could improve model generalization.

Validate across sites - Test the approach on data from other PV installation sites to see if accuracy gains hold in different locations/weather patterns.

Compare to ensemble benchmark - Include a well-optimized ensemble as a benchmark, not just other individual models, to fully justify the two-stage approach.  

Conclusions

The conclusions drawn in the paper are consistent with the evidence and arguments presented and do adequately address the main research question.

Furthermore, it is suggested to incorporate a section titled "Suggestions for Future Research" to enhance the study's scientific value. Overall, the manuscript is well-presented, organized, and the methodology is acceptable and engaging, making it suitable for publication after these minor revisions.

Author Response

(The authors gave the same response as above.)

Reviewer 4 Report

Comments and Suggestions for Authors

Fascinating and timely article. It deserves publication, and I am recommending acceptance with corrections. Some issues require your attention. I list these corrections below as feedback/comments, and I look forward to reading this article's updated version. 

The level of originality of the paper is high. The literature review and proposed design are suitable. The results are discussed: authors found the values and risks of new short-term photovoltaic voltage (PV) prediction scheme.

You have done a really good job at reviewing so many articles, but you didn’t discuss new and emerging forms of data and technologies, which seems to be a new field for future developments in IoT data and energy use. There are recent articles on this topic that review recent and relevant literature, for example, on the related topic of ‘New and emerging forms of data and technologies’ - see: https://doi.org/10.1007/s11042-022-13451-5 It would be interesting to read your thoughts on this emerging area of risk, and given that its coming in the future, your article would stand strong with time if you include a short discussion on this topic. I would also be very interested to see a few sentences reviewing and comparing your work in relation to this very recent study in a related topic.

One final comment: you should check if all the things discussed in the introduction are also discussed in the conclusion. because the introduction is much longer than the conclusion. Usually, these sections are comparable in length. If you think you have covered everything, that’s OK, but to mention that conclusion is the best chapter to outline your key findings and key conclusions. So, it would be best if you used this chapter to make your article more readable, and since most readers would focus a great deal of their attention on the conclusion, this section should make the key conclusions more visible (and hence more interesting).  

I hope the comments and feedback are helpful, and well done for writing such an interesting article. I am looking forward to reading the updated version.

Comments on the Quality of English Language

Language seems OK. 

Author Response

(The authors gave the same response as above.)

Round 2

Reviewer 1 Report

Comments and Suggestions for Authors

Accept in present form.